# Effect of Hydroxyl Groups Esterification with Fatty Acids on the Cytotoxicity and Antioxidant Activity of Flavones

**DOI:** 10.3390/molecules27020420

**Published:** 2022-01-10

**Authors:** Grażyna Kubiak-Tomaszewska, Piotr Roszkowski, Emilia Grosicka-Maciąg, Paulina Strzyga-Łach, Marta Struga

**Affiliations:** 1Department of Biochemistry and Pharmacogenomics, Faculty of Pharmacy, Medical University of Warsaw, Banacha 1, 02-097 Warszawa, Poland; grazyna.kubiak-tomaszewska@wum.edu.pl; 2Faculty of Chemistry, University of Warsaw, Pasteura 1, 02-093 Warsaw, Poland; 3Chair and Department of Biochemistry, Medical University of Warsaw, Banacha 1, 02-097 Warszawa, Poland; paulina.strzyga@wum.edu.pl (P.S.-Ł.); marta.struga@wum.edu.pl (M.S.)

**Keywords:** flavonoids, fatty acids conjugates, lipid peroxidation, *MDA*, antioxidant potential

## Abstract

Flavonoids and polyunsaturated fatty acids due to low cytotoxicity in vitro studies are suggested as potential substances in the prevention of diseases associated with oxidative stress. We examined novel 6-hydroxy-flavanone and 7-hydroxy-flavone conjugates with selected fatty acids (FA) of different length and saturation and examined their cytotoxic and antioxidant potential. Our findings indicate that the conjugation with FA affects the biological activity of both the original flavonoids. The conjugation of 6-hydroxy-flavanone increased its cytotoxicity towards prostate cancer PC3 cells. The most noticeable effect was found for oleate conjugate. A similar trend was observed for 7-hydroxy-flavone conjugates with the most evident effect for oleate and stearate. The cytotoxic potential of all tested conjugates was not specific towards PC3 because the viability of human keratinocytes HaCaT cells decreased after exposure to all conjugates. Additionally, we showed that esterification of the two flavonoids decreased their antioxidant activity compared to that of the original compounds. Of all the tested compounds, only 6-sorbic flavanone showed a slight increase in antioxidant potential compared to that of the original compound. Our data show that conjugated flavonoids are better absorbed and enhance cytotoxic effects, but the presence of FA lowered the antioxidant potential.

## 1. Introduction

Although ROS are a product of cells metabolism, they may cause oxidative damage when produced in excessive quantities. They can interact with nucleic acids, proteins and lipids leading to severe metabolic disturbances. ROS have been implicated in the pathogenesis of a wide variety of diseases, including cancer, neurodegenerative diseases, metabolic diseases, cardiovascular diseases, respiratory diseases or rheumatoid arthritis [1,2,3,4].

The physiological production of ROS during aerobic metabolism requires the presence of an effective antioxidant system. A weak defense system facilitates the formation of both superoxide anions and hydrogen peroxide. Antioxidants play a major role in protection against molecular oxidative damage and may have therapeutical relevance in oxidative stress-mediated diseases. Therefore, they should be present in high concentration when a large quantity of ROS are formed [5].

Flavonoids are widely distributed in fruits, vegetables, grains, nuts, seeds, tea and traditional medicinal herbs supplements and many of them are components of the human daily diet. They are known to be antioxidants that can protect the cell from oxidative stress. Science reports indicate that although these natural compounds are considered as a nonessential dietary factor, they are a very important link between the diet and prevention of many diseases, including cardiovascular and cancer. Numerous in vitro studies have described anticancer activity of flavonoids in different cancer cells lines; additionally, they indicate flavonoids as a potential substances in cancer prevention [6,7,8].

The basic structure of flavonoids (flavones) consists of three rings involving A and B (benzene rings) and C (heterocyclic pyran ring) [9,10]. They are classified into different subgroups based on the carbon of the C ring on which the B ring is linked and the degree of unsaturation and oxidation of the C ring. The flavones and flavanones are two very important classes of flavonoids. They differ in the presence of saturated double bond between position 2 and 3 in C ring of flavanone molecule. The occurrence of numerous hydroxyl groups in B and/or A rings facilitates scavenging of hydroxyl radicals and inhibit lipid peroxidation. Moreover, hydroxyl groups can chelate pro-oxidative metal ions including Fe^2+^ or Cu^2+^ [11,12]. In turn, the presence of hydroxyl groups in C4 and/or C7 positions determines their ability to interact with the estrogen receptor [13]. Flavonoids as inhibitors of cytochrome P450 isoenzymes also prevent the activation of numerous procarcinogen compounds [9]. Additionally, the modulating effect of flavonoids on angiogenesis, adipogenesis and inflammatory processes has been also reported [14,15,16]. Moreover, these compounds may interact directly with proteins, lipoproteins and nucleic acids [17]. Such diversified action makes flavonoids useful as potential drugs, including the treatment of cancer, cardiovascular diseases, neurodegenerative diseases, viral diseases and hormone replacement therapy in menopausal women [18,19,20,21,22]. However, it should be stressed that flavonoids are not still evaluated for their safety profile and tumor site-specific action.

Although most flavonoids reveal high antioxidant potential, they may also exhibit pro-oxidative properties. This type of action is characteristic of polyphenolic compounds containing a catechol group that can form semioquinone radicals (e.g., quercetin belonging to the group of flavonols). Pro-oxidative properties were also observed in in vitro studies in the presence of transition metal ions, including Cu^2+^. Moreover, the presence of the Δ^2,3^ double bond promotes the pro-oxidative effect of flavonoids [23,24,25].

Flavonoids are consumed with diet in the form of aglycons or glycosides. Individual forms are transported to enterocytes by a passive (aglycons) or active (glycosides) transport [26,27,28]. Their transport across biological membranes and the bioavailability are significantly limited because of their large size. Unabsorbed glycoside derivatives are hydrolyzed by bacterial β-lactamases and are absorbed in the large intestine as an aglycone form. Biotransformation includes glucuronidation of aglycons in enterocytes and hepatic hydroxylation, demethylation, O-methylation or conjugation with active sulfuric acid [29,30,31]. The type and structural position of conjugation can significantly affect the antioxidant activity of flavonoids, time elimination from the bloodstream and their toxicity [32]. Therefore, new flavonoid derivatives with high antioxidant activity, better bioavailability and lower possible toxicity are being investigated.

The tested compounds 6-hydroxy-flavanone and 7-hydroxy-flavone belong to the group of hydroxylated flavones with proven antioxidant, anti-inflammatory, antimicrobial and antitumor properties, which are not only ROS scavengers, but can also induce cell death through apoptosis or necrosis by inhibition of efflux pumps [33,34]. Therefore, it can be expected that their conjugation with fatty acids, especially PUFAS being agents with antioxidant anti-angiogenic activity, will influence their activity.

Based on these findings, we have synthesized the series of the novel conjugates of 6-hydroxy-flavanone and 7-hydroxy-flavone with fatty acids of different length and saturation. The new derivatives were evaluated for their cytotoxic activity in metastatic cancer cells and normal cells. Additionally, the studied compounds were tested on the antioxidant properties.

## 2. Results and Discussion

### 2.1. Chemistry

In this study we synthesized esters of selected fatty acids 6-hydroxy-flavanone and 7-hydroxy-flavone. The fatty acids, including sorbic, stearic, oleic, linoleic and linolenic, were transformed to appropriate acyl chlorides in reaction with oxalyl chloride and were used without further purification. In the next step, the reaction between 6-hydroxy-flavanone and 7-hydroxy-flavone and freshly prepared acyl chloride produced a series of esters. The esterification of phenol function was carried out in the presence of pyridine, as shown in Figure 1.

### 2.2. Viability Test

Recently, increasing emphasis in modern cancer therapy is imposed on looking for drugs that show a mild effect on healthy normal cells with a simultaneous high activity in relation to cancer cells. Flavonoids belongs to such a group of compounds demonstrating a high cytotoxic potential toward various human cancer cells with little or no effect on normal cells. This fact has stimulated large interest in developing potential flavonoid-based chemotherapeutics for anticancer treatment [33,34,35,36]. The biological activity of flavonoids on cell functions mainly depends on their chemical structure. Cytotoxicity and/or antiproliferative potential of flavonoids is conditioned by different structural factors, including the saturation of the C2-C3 bond and the position of both the number and substitution of hydroxyl groups in the A and B rings [32]. It is known that even a slight alternation in the chemical structures affects their activity and flavonoids with very similar chemical structures do not exhibit the identical biological potential [6,37,38]. Therefore, advanced research involving flavonoid derivatives with a high antioxidant activity, and the better bioavailability along the lower toxicity are included. Consequently, we have synthesized novel conjugates of 6-hydroxy-flavanone and 7-hydroxy-flavone with fatty acids with different length and saturation. Polyunsaturated fatty acids (PUFAS) except for flavonoids are also groups of compounds that have attracted worldwide scientific interest in the past 30 years. They are both cardiovascular protective and anti - inflammatory agents. Sun et al. demonstrates that flavonoid fatty acid conjugates have at least similar or higher antioxidant, anti-peroxidation and anti-angiogenic activity [39].

The cytotoxic effect of 6-hydroxy-flavanone and 7-hydroxy-flavone and their conjugates was evaluated by MTT dye reduction assay, measuring mitochondrial respiratory functions. We have chosen two types of cell culture lines: human prostate cancer cells (PC3) and normal human immortal keratinocyte cells (HaCaT) (Table 1).

6-hydroxy-flavanone is a synthetic compound and belongs to the group of flavanones. Several studies showed that naturally occurring flavanone compounds beside of antioxidant activity exhibit cytotoxic action towards some types of cancer cells, including leukemia cells (TPH1, U937,) colon cancer cell (SNUC4) and hepatocarcinoma SMM7721 cells [40]. To assess the cytotoxic potential of tested compounds, we incubated both PC3 and HaCat cells with 6-hydroxy-flavanone or 7-hydroxy-flavon and all newly synthesized conjugates during 72 h. 6-hydroxy-flavanone at tested concentrations (0–100 μM) did not exhibit high cytotoxic activity towards both HaCaT and PC3 cells after 72 h incubation, 100 μM concentration reduced cells viability to 73% and 89% respectively compared to that in the control cells (IC_50_ > 100 μM). On the other hand, the conjugation with fatty acids (except of stearic acid) slightly increases the cytotoxic potential of 6-hydroxy-flavanone towards PC3 cells. Our results are similar to those published by Szliszka et al. [40]. They noticed that 6-hydroxy-flavanone conjugated with palmitic acid did not exhibit anticancer effects, whereas conjugation with propionic acids increased cytotoxic action towards HeLa cells in a dose dependent manner (25–100 μM) [40]. The highest cytotoxic potency on PC3 cells was observed for monounsaturated oleic acid conjugate (IC_50_ = 38.1 ± 3.24 μM). Polyunsaturated conjugates, including linolate, linolenate and sorbate exhibited similar cytotoxic potential on PC3 cells: IC_50_ = 48.32 ± 8.73 μM, IC_50_ = 51.4 ±.2.8 μM and IC_50_ = 54.7 ± 5.25 μM. It is worth emphasizing that conjugation of 6-hydoxy-flavanone with FA increased its cytotoxic potential also toward normal HaCaT cells. Saturated stearate conjugate did not show cytotoxic action toward PC3 cells (IC_50_ > 100 μM) but it revealed a significant cytotoxic potential against normal HaCaT cells (IC_50_ = 40.61 ± 4.21 μM). The similar values of IC_50_ were obtained for both linolenate (IC_50_ = 44.6 ± 5.89 μM) and oleate acid (IC_50_ = 53.29 ± 6.87 μM). The lowest cytotoxic potentials against HaCaT cells IC_50_ = 79.8 ± 2.21 μM and IC_50_ = 80 ± 4.75 μM was proved for sorbate and linolenate conjugates, respectively. 7-hydroxy-flavone similarly to 6-hydroxy-flavonone did not exhibit cytotoxic activity against HaCaT cells, on the other site it reduced PC3 cells viability by 38% compared to that in control cells. All examined conjugates (except for 7-linolenate IC_50_ > 100 μM and 7-sorbate acid IC_50_ = 94.3 ± 4.24 μM) of 7-hydroxy-flavone exhibit very similar anticancer potential (IC_50_ = 33.2 ± 4.81 μM–41.2 ± 2.2 μM) on PC3 cells. The strongest effect was observed for stearate conjugate. The obtained results suggest that the cytotoxic potential of stearate and linolenate conjugates in cancer cells may depend on the esterification position. We observed the most pronounced cytotoxic effect on PC3 cells for saturated 7-stearate conjugate and at the same time the 6-stearate conjugate was neutral for these cells. On the other hand, 7 linolenate conjugate did not exhibit cytotoxic properties, whereas 6 linolenate was more toxic in PC3 cells. Simultaneously, all tested 7 hydroxy conjugate were also more active toward HaCaT cells (IC_50_ = 44.9 ± 6.44 – 65 ± 4.12 μM). However, limited research into synthetic fatty acid and flavonoid conjugates and their biological activity has been performed until now. The presented results suggest that cytotoxic activity of both examined flavonoids increases after conjugation with fatty acids. Yang et al. proved that some fatty acids modulate the lipid structure of cellular membranes and consequently facilitating the binding and activation of several important signaling proteins [41]. We could also suppose that conjugation with fatty acids increases the hydrophobic properties of flavonoids and facilitates their cellular absorption. Our results are similar to those obtained by Soufi et al. [42]. They noticed that conjugation of natural flavonoids quercetin with omega 3 and omega 6 fatty acids increased the cytotoxic action compared to un-esterified form in MCF-7 cells. Additionally, the research published by Denihelova et al. revealed that conjugation of quercetin with acyl groups increased its lipophilic properties and improved cytotoxic action towards cancer and normal cells [43].

According to the obtained data, we can conclude that the cytotoxic potential of the tested flavonoid derivatives is not specific towards prostate cancer cells because they also decrease the viability of control normal cells. It may be also considered that the cytotoxic activity of all examined compounds strongly depends on hydroxyl group position and type of conjugated fatty acid.

### 2.3. Antioxidant Activity

It is well known that excessive and prolonged oxidative stress may cause metabolic disorders. Numerous reports indicate that the toxic oxygen derivatives generated during oxidative stress damage macromolecules, including proteins, lipids and nucleic acids [44,45,46,47]. The polyunsaturated fatty acids, which are components of membrane phospholipids (mainly phosphatidylethanolamine and phosphatidylcholine), are the most susceptible to peroxidation. The peroxidation reactions display an avalanche and free radical character and leads to lipid peroxide production. Malondialdehyde (*MDA*) and trans-4-hydroxynonenal (4HNE) that may damage nucleic acid molecules and can be lipid peroxidation markers are the most important end products of the reactions. 4HNE is the most toxic and active product, which may form covalent adducts with nucleophilic groups in proteins, nucleic acids and membrane lipids, thus affecting many cellular signaling pathways. On the other hand, *MDA* is a bacterial and mammalian cell mutagen and may be carcinogenic to rats. The two compounds have a longer half-life than ROS and diffuse easily in the cell [48,49,50,51].

Both enzymatic antioxidant defense, including catalase (CAT), glutathione peroxidase (GPX), superoxide dismutase (SOD), the thioredoxin system and non-enzymatic antioxidants glutathione participate in the inactivation of ROS and lipid peroxides [52,53,54,55]. During conditions of extremely high or chronic oxidative stress, the endogenous antioxidant system can be insufficient, and there is a need to support cellular systems with exogenous antioxidants. It is well known that prolonged and chronic oxidative stress is implicated in the development of many diseases. Considering that, numerous reports describe flavonoids as a promising alternative to conventional therapy for various oxidative stress-related diseases [56].

Our research shows that the cytotoxic action of all examined flavonoid conjugates dependend on hydroxyl group position and substitution. Additionally, Verma at al. [57] indicate that biological activity of flavonoids is related to the nature and position of substituents in their ring system. Moreover, results presented by Masek at al. [58] prove that antioxidant properties of flavonoids strongly depend on hydroxyl group position.

The antioxidant potential of the new flavone derivatives was assessed based on the analysis of malondialdehyde concentration formed as a product of lipid oxidation stimulated by iron (II) ions [59]. The results are presented in Table 2.

Our research shows that both 6-hydroxy-flavanone and 7-hydroxy-flavone exhibit slight antioxidant activities and reduced the level of *MDA* as compared to that in the control sample by 26.62% and 20.14%, respectively. The obtained results confirmed the thesis presented by Masek et al. [58]. They demonstrated that hydroxyl groups at the R-6 and R-7 positions in ring A of flavonoids do not significantly affect the ability to scavenge ROS and the antioxidant activity of the two tested compounds was comparable to the product with no OH groups in the structure. On the other hand, Wang et al. showed that the substitution of A-ring by donating electrons OH groups at these positions enhances the inhibitory effect on nitric oxide synthase as a source of NO, a substrate for the synthesis of reactive nitrogen species. This effect was attributed to the increased possibility of electron donation by the ring A resonance structure in favor of the oxygen atom of the benzopyrone ring [60]. Moreover, it is known that hydroxylation of the A ring at the positions 6 and 7 enhances the inhibitory effect of Akt phosphorylation and prevents ROS-induced autophagy. Our research showed that esterification of the OH groups in position 6 and 7 with polyunsaturated fatty acids slightly decreased the antioxidant potential of the original compounds. The presented results are in agreement with some published data. They indicate that esterification/blocking of hydroxyl groups in the flavonoids structure decrease the ability to scavenge free radicals or chelate metals [43,61,62]. Therefore, it may be assumed that the inclusion of a component susceptible to oxidation in the flavone structure (the possibility of creating a lipid radical or lipid peroxy radical) results in a reduction of the antioxidant potential of flavones. Moreover, the reduction in the antioxidant activity of the tested hydroxy derivatives of flavones after esterification of the hydroxyl groups in the 6 and 7 positions of the A ring may be explained by the abolition of the possibility of donating reducing equivalents to oxygen radicals by these groups.

Our result show that only when in conjunction with sorbic acid, there was an increase in the antioxidant activity of 6-hydrox-flavanone. Sorbic acid is an antibacterial and antifungal compound used in food protection. This short-chain unsaturated fatty acid penetrates cellular membranes rather easily, which may favor the transport of conjugated flavonoids. The inhibitory effect of sorbic acid on the production of F2-isoprostanes, which is an indicator of lipid peroxidation, has also been demonstrated [63,64,65]. This is especially important in the case of actively metabolic organs, including the brain. High iron content and high oxygen consumption of this organ results in an increased risk of oxygen radical formation. Our findings seem to confirm the protective effect of sorbic acid on rat brain cells under the conditions of oxidative stress induced by Fe(II) ions. At a high degree of dissociation, sorbic acid increases the pool of H^+^ ions in cells and provides substrates necessary for the inactivation of ROS. Moreover, sorbic acid has an inhibitory effect on neutrophil myeloperoxidase, which reduces oxidative stress [65]. This may explain the higher activity of 6-sorbic flavanone. This beneficial effect of sorbic acid is probably related to the characteristic chemical configuration of this conjugate. Either the double bond system on the resonance structure of ring A of 6-hydroxy-flavanone or the 4-carbonyl group and the C2 = C3 double bond may significantly increase the antioxidant activity of these compounds. Changing the position of the substituent from 6 (**4a**) to 7 (**5a**) may cause a difference in the interaction of the sorbic acid chain with the electrons of the aromatic flavone ring. Such a change may result in differences in the biological activity of the derivative **4a** and **5a**.

## 3. Materials and Methods

### 3.1. Apparatus, Materials, and Analysis

Dichloromethane, tetrahydrofuran and methanol were supplied from Sigma Aldrich (Saint Louis, MO, USA). The sorbic acid (99%) was purchased from Alfa Aesar (Haverhill, MA, USA) company, stearic acid (95%), oleic acid (>99%), linoleic acid (≥99%), linolenic acid (≥99%), (±) 6-hydroxy-flavanone (99%), 7-hydroxy-flavone (≥98%), pyridine (≥99%) and oxalyl chloride (98%) were purchased from Sigma Aldrich (Saint Louis, MO, USA). All other chemicals were of analytical grade and were used without any further purification. The NMR spectra were recorded on a Bruker AVANCE (Bruker, Karlsruhe, Germany) spectrometer operating at 500 MHz for 1H NMR and at 125 MHz for 13C NMR. The spectra were measured in CDCl3 and are given as d values (in ppm) relative to TMS. Mass spectral ESI measurements were carried out on LCT Micromass TOF HiRes apparatus (Micromass UK Limited, Manchester, UK). Melting points were determined on a Melting Point Meter KSP1D (A. Krüss Optronic, Hamburg, Germany) and were uncorrected. TLC analyses were performed on silica gel plates (Merck Kiesegel GF254, Merck, Darmstadt, Germany) and visualized using UV light or iodine vapor. Column chromatography was carried out at atmospheric pressure using Silica Gel 60 (230–400 mesh, Merck, Darmstadt, Germany) using appropriate eluents. Tris-HCl, thiobarbituric acid, FeCl2, ascorbic acid, 1-(4,5-Dimethylthiazol-2-yl)-3,5-diphenylformazan, DMSO, isopropanol and SDS were purchased from Sigma Aldrich (Saint Louis, MO, USA).

### 3.2. The Esters of 6-Hydroxy-flavanone and 7-Hydroxy-flavone

#### 3.2.1. General Procedure for 6-Hydroxy-flavanone Esters of (3, 5, 7, 9, 11) Synthesis

To a magnetically stirred at 22–23 °C solution of (±) 6-hydroxy-flavanone (0.24 g; 1.0 mmol) in dichloromethane:tetrahydrofuran mixture (10 mL:5 mL), pyridine (0.20 mL; 2.5 mmol) and next a solution of acyl chloride (2.0 mmol) in 2 mL of dichloromethane were added. The resulting solution was stirred at 22–23 °C for 16 h and concentrated under reduced pressure. The residue was dissolved in CH_2_Cl_2_ (40 mL) and water (10 mL) and 3% HCl_aq_ solution (to pH = 2–3) were added. After separation of the phases, the water layer was extracted with CH_2_Cl_2_ (20 mL). The combined organic layers were washed with 3% HCl_aq_ solution (10 mL) and distilled water (15 mL). The organic layer was dried over MgSO_4_ and after evaporation of the solvent under reduced pressure the product was isolated using column chromatography on silica gel and CH_2_Cl_2_: MeOH mixture (0–1% MeOH) as an eluent. (Appendix A).

#### 3.2.2. General procedure for Acyl Chlorides Synthesis

To a magnetically stirred at 2–5 °C solution of carboxylic acid (8.92 mmol) in dichloromethane (10 mL), oxalyl chloride (17.80 mmol) was added and the solution was stirred for 3 h. The evaporation of the solvent and excess of oxalyl chloride gave quantitatively acyl chloride, which was used without further purification


**6-hydroxy-flavanone sorbic ester (4a)**


Pale yellow solid, 240 mg (72%). Mp. = 129.4–130.8 °C.

^1^H NMR (CDCl_3_, 500 MHz) δ (ppm): 1.89 (d, *J* = 6.0 Hz, 3*H*), 2.88 (dd, *J*_1_ = 17.0 Hz, *J*_2_ = 3.0 Hz, 1*H*), 3.07 (dd, *J*_1_ = 17.0 Hz, *J*_2_ = 13.5 Hz, 1*H*), 5.47 (dd, *J*_1_ = 13.5 Hz, *J*_2_ = 2.5 Hz, 1*H*), 5.95 (d, *J* = 15.0 Hz, 1*H*), 6.20–6.31 (m, 2*H*), 7.06 (d, *J* = 8.5 Hz, 1*H*), 7.27 (dd, *J*_1_ = 8.5 Hz, *J*_2_ = 2.5 Hz, 1*H*), 7.38–7.48 (m, 6*H*), 7.66 (d, *J* = 3.0 Hz, 1*H*). ^13^C NMR (CDCl_3_, 125 MHz) δ (ppm): 18.8, 44.4, 79.8, 117.5, 119.1, 119.2, 121.1, 126.1 (2xC), 128.8, 128.8 (2xC), 129.7, 130.0, 138.5, 141.0, 144.9, 147.3, 159.0, 165.7, 191.2.

HR-MS (ESI) *m/z* 357.1115 (calcd for C_21_H_18_O_4_Na [M + Na]^+^, 357.1103).


**7-hydroxy-flavone sorbic ester (5a)**


White solid, 265 mg (80%). Mp. = 137.8–139.7 °C.

^1^H NMR (CDCl_3_, 500 MHz) δ (ppm): 1.91 (d, *J* = 5.0 Hz, 3*H*), 5.98 (d, *J* = 15.0 Hz, 1*H*), 6.25–6.33 (m, 2*H*), 6.80 (s, 1*H*), 7.19 (dd, *J*_1_ = 9.0 Hz, *J*_2_ = 2.5 Hz, 1*H*), 7.45 (d, *J* = 2.5 Hz, 1*H*), 7.46–7.53 (m, 4*H*), 7.87–7.90 (m, 2*H*), 8.24 (d, *J* = 9.0 Hz, 1*H*). ^13^C NMR (CDCl_3_, 125 MHz) δ (ppm): 18.8, 107.5, 111.0, 117.1, 119.4, 121.5, 126.2 (2xC), 126.9, 129.0 (2xC), 129.6, 131.5, 131.6, 141.8, 148.0, 154.8, 156.6, 163.5, 164.7, 177.6.

HR-MS (ESI) *m/z* 355.0935 (calcd for C_21_H_16_O_4_Na [M + Na]^+^, 355.0946).


**6-hydroxy-flavanone stearic ester (4b)**


White solid, 450 mg (88%). Mp. = 61.2–62.4 °C.

^1^H NMR (CDCl_3_, 500 MHz) δ (ppm): 0.88 (t, *J* = 7.0 Hz, 3*H*), 1.26–1.35 (m, 26*H*), 1.38–1.48 (m, 2*H*), 1.72–1.78 (m, 2*H*), 2.55 (t, *J* = 7.5 Hz, 2*H*), 2.89 (dd, *J*_1_ = 17.0 Hz, *J*_2_ = 3.0 Hz, 1*H*), 3.07 (dd, *J*_1_ = 17.0 Hz, *J*_2_ = 13.0 Hz, 1*H*), 5.48 (dd, *J*_1_ = 13.5 Hz, *J*_2_ = 3.0 Hz, 1*H*), 7.06 (d, *J* = 9.0 Hz, 1*H*), 7.22 (dd, *J*_1_ = 9.0 Hz, *J*_2_ = 3.0 Hz, 1*H*), 7.39–7.49 (m, 5*H*), 7.61 (d, *J* = 3.0 Hz, 1*H*). ^13^C NMR (CDCl_3_, 125 MHz) δ (ppm): 14.1, 22.7, 24.9, 29.1, 29.3, 29.4, 29.5, 29.6, 29.7, 29.7, 29.7, 29.7 (4xC), 31.9, 34.3, 44.4, 79.8, 119.2, 119.2, 121.2, 126.1 (2xC), 128.9 (3xC), 129.9, 138.5, 144.9, 159.1, 172.4, 191.2.

HR-MS (ESI) *m/z* 529.3281 (calcd for C_33_H_46_O_4_Na [M + Na]^+^, 529.3294).


**7-hydroxy-flavone stearic ester (5b)**


White solid, 410 mg (81%). Mp. = 84.5–85.7 °C.

^1^H NMR (CDCl_3_, 500 MHz) δ (ppm): 0.88 (t, *J* = 7.0 Hz, 3*H*), 1.26–1.37 (m, 26*H*), 1.41–1.46 (m, 2*H*), 1.75–1.81 (m, 2*H*), 2.61 (t, *J* = 7.5 Hz, 2*H*), 6.80 (s, 1*H*), 7.15 (dd, *J*_1_ = 9.0 Hz, *J*_2_ = 2.5 Hz, 1*H*), 7.41 (d, *J* = 2.5 Hz, 1*H*), 7.49–7.55 (m, 3*H*), 7.89–7.91 (m, 2*H*), 8.24 (d, *J* = 9.0 Hz, 1*H*). ^13^C NMR (CDCl_3_, 125 MHz) δ (ppm): 14.1, 22.7, 24.8, 29.1, 29.2, 29.4, 29.4, 29.6, 29.6, 29.7, 29.7, 29.7 (4xC), 31.9, 34.4, 107.6, 111.0, 119.4, 121.6, 126.2 (2xC), 127.0, 129.0 (2xC), 131.5, 131.7, 154.7, 156.7, 163.6, 171.4, 177.7.

HR-MS (ESI) *m/z* 527.3146 (calcd for C_33_H_44_O_4_Na [M + Na]^+^, 527.3137).


**6-hydroxy-flavanone oleic ester (4c)**


White solid, 460 mg (91%). Mp. = 46.2–47.1 °C.

^1^H NMR (CDCl_3_, 500 MHz) δ (ppm): 0.88 (t, *J* = 7.0 Hz, 3*H*), 1.27–1.37 (m, 18*H*), 1.39–1.448 (m, 2*H*), 1.72–1.78 (m, 2*H*), 2.00–2.05 (m, 4*H*), 2.55 (t, *J* = 7.5 Hz, 2*H*), 2.89 (dd, *J*_1_ = 17.0 Hz, *J*_2_ = 3.0 Hz, 1*H*), 3.07 (dd, *J*_1_ = 17.0 Hz, *J*_2_ = 13.5 Hz, 1*H*), 5.32–5.39 (m, 2*H*), 5.48 (dd, *J*_1_ = 13.5 Hz, *J*_2_ = 3.0 Hz, 1*H*), 7.07 (d, *J* = 9.0 Hz, 1*H*), 7.23 (dd, *J*_1_ = 8.5 Hz, *J*_2_ = 2.5 Hz, 1*H*), 7.39–7.48 (m, 5*H*), 7.61 (d, *J* = 3.0 Hz, 1*H*). ^13^C NMR (CDCl_3_, 125 MHz) δ (ppm): 14.1, 22.7, 24.9, 27.2, 27.2, 29.1 (2xC), 29.2, 29.3, 29.3, 29.5, 29.7, 29.8, 31.9, 34.2, 44.4, 79.8, 119.2, 119.2, 121.2, 126.1 (2xC), 128.9, 128.9 (2xC), 129.7, 129.9, 130.0, 138.5, 144.8, 159.1, 172.4, 191.2.

HR-MS (ESI) *m/z* 527.3128 (calcd for C_33_H_44_O_4_Na [M + Na]^+^, 527.3137).


**7-hydroxy-flavone oleic ester (5c)**


White solid, 400 mg (80%). Mp. = 73.3–74.5 °C.

^1^H NMR (CDCl_3_, 500 MHz) δ (ppm): 0.88 (t, *J* = 7.0 Hz, 3*H*), 1.27–1.38 (m, 18*H*), 1.41–1.47 (m, 2*H*), 1.76–1.82 (m, 2*H*), 2.00–2.05 (m, 4*H*), 2.62 (t, *J* = 7.5 Hz, 2*H*), 5.33–5.39 (m, 2*H*), 6.81 (s, 1*H*), 7.15 (dd, *J*_1_ = 8.5 Hz, *J*_2_ = 2.0 Hz, 1*H*), 7.41 (d, *J* = 2.0 Hz, 1*H*), 7.50–7.56 (m, 3*H*), 7.89–7.91 (m, 2*H*), 8.24 (d, *J* = 9.0 Hz, 1*H*). ^13^C NMR (CDCl_3_, 125 MHz) δ (ppm): 14.1, 22.7, 24.8, 27.2, 27.2, 29.1, 29.1, 29.2, 29.3, 29.3, 29.5, 29.7, 29.8, 31.9, 34.4, 107.6, 111.0, 119.4, 121.7, 126.2 (2xC), 127.1, 129.1 (2xC), 129.7, 130.1, 131.5, 131.7, 154.7, 156.7, 163.6, 171.4, 177.6.

HR-MS (ESI) *m/z* 525.2995 (calcd for C_33_H_42_O_4_Na [M + Na]^+^, 525.2981).


**6-hydroxy-flavanone linoleic ester (4d)**


Colorless oil, 450 mg (89%).

^1^H NMR (CDCl_3_, 500 MHz) δ (ppm): 0.89 (t, *J* = 7.0 Hz, 3*H*), 1.27–1.43 (m, 14*H*), 1.72–1.78 (m, 2*H*), 2.03–2.09 (m, 4*H*), 2.55 (t, *J* = 7.5 Hz, 2*H*), 2.77–2.80 (m, 2*H*), 2.89 (dd, *J*_1_ = 17.0 Hz, *J*_2_ = 3.0 Hz, 1*H*), 3.07 (dd, *J*_1_ = 17.0 Hz, *J*_2_ = 13.5 Hz, 1*H*), 5.31–5.41 (m, 4*H*), 5.48 (dd, *J*_1_ = 13.5 Hz, *J*_2_ = 3.0 Hz, 1*H*), 7.07 (d, *J* = 9.0 Hz, 1*H*), 7.23 (dd, *J*_1_ = 9.0 Hz, *J*_2_ = 2.0 Hz, 1*H*), 7.38–7.49 (m, 5*H*), 7.61 (d, *J* = 2.5 Hz, 1*H*). ^13^C NMR (CDCl_3_, 125 MHz) δ (ppm): 14.1, 22.6, 24.9, 25.6, 27.2, 27.2, 29.1, 29.1, 29.2, 29.4, 29.6, 31.5, 34.2, 44.4, 79.8, 119.2, 119.2, 121.2, 126.1 (2xC), 127.9, 128.1, 128.9, 128.9 (2xC), 129.9, 130.0, 130.2, 138.5, 144.8, 159.1, 172.4, 191.2.

HR-MS (ESI) *m/z* 525.2990 (calcd for C_33_H_42_O_4_Na [M + Na]^+^, 525.2981).


**7-hydroxy-flavone linoleic ester (5d)**


Colorless solidifying oil, 410 mg (86%).

^1^H NMR (CDCl_3_, 500 MHz) δ (ppm): 0.88 (t, *J* = 7.0 Hz, 3*H*), 1.28–1.46 (m, 14*H*), 1.76–1.82 (m, 2*H*), 2.03–2.08 (m, 4*H*), 2.62 (t, *J* = 7.5 Hz, 2*H*), 2.77–2.80 (m, 2*H*), 5.31–5.41 (m, 4*H*), 6.81 (s, 1*H*), 7.15 (dd, *J*_1_ = 9.0 Hz, *J*_2_ = 2.5 Hz, 1*H*), 7.41 (d, *J* = 2.0 Hz, 1*H*), 7.50–7.56 (m, 3*H*), 7.89–7.91 (m, 2*H*), 8.24 (d, *J* = 9.0 Hz, 1*H*). ^13^C NMR (CDCl_3_, 125 MHz) δ (ppm): 14.1, 22.6, 24.8, 25.6, 27.2, 27.2, 29.0, 29.1, 29.1, 29.3, 29.6, 31.5, 34.4, 107.6, 111.0, 119.4, 121.7, 126.2 (2xC), 127.0, 127.8, 128.1, 129.0 (2xC), 129.9, 130.2, 131.5, 131.7, 154.7, 156.7, 163.6, 171.4, 177.6.

HR-MS (ESI) *m/z* 523.2837 (calcd for C_33_H_40_O_4_Na [M + Na]^+^, 523.2824).


**6-hydroxy-flavanone linolenic ester (4e)**


Colorless oil, 450 mg (90%).

^1^H NMR (CDCl_3_, 500 MHz) δ (ppm): 0.98 (t, *J* = 7.5 Hz, 3*H*), 1.31–1.43 (m, 8*H*), 1.72–1.78 (m, 2*H*), 2.01–2.01 (m, 4*H*), 2.55 (t, *J* = 7.5 Hz, 2*H*), 2.75–2.83 (m, 4*H*), 2.89 (dd, *J*_1_ = 17.0 Hz, *J*_2_ = 3.0 Hz, 1*H*), 3.07 (dd, *J*_1_ = 17.0 Hz, *J*_2_ = 13.5 Hz, 1*H*), 5.29–5.43 (m, 6*H*), 5.48 (dd, *J*_1_ = 13.5 Hz, *J*_2_ = 3.0 Hz, 1*H*), 7.06 (d, *J* = 9.0 Hz, 1*H*), 7.22 (dd, *J*_1_ = 9.0 Hz, *J*_2_ = 3.0 Hz, 1*H*), 7.37–7.48 (m, 5*H*), 7.61 (d, *J* = 3.0 Hz, 1*H*). ^13^C NMR (CDCl_3_, 125 MHz) δ (ppm): 14.3, 20.6, 24.9, 25.5, 25.6, 27.2, 29.1, 29.1, 29.1, 29.6, 34.2, 44.4, 79.8, 119.1, 119.2, 121.1, 126.1 (2xC), 127.1, 127.8, 128.2, 128.3, 128.8, 128.9 (2xC), 129.9, 130.2, 131.9, 138.5, 144.8, 159.1, 172.3, 191.2.

HR-MS (ESI) *m/z* 523.2834 (calcd for C_33_H_40_O_4_Na [M + Na]^+^, 523.2824).


**7-hydroxy-flavone linolenic ester (5e)**


Pale yellow oil, 390 mg (78%).

^1^H NMR (CDCl_3_, 500 MHz) δ (ppm): 0.98 (t, *J* = 7.5 Hz, 3*H*), 1.37–1.46 (m, 8*H*), 1.76–1.82 (m, 2*H*), 1.99–2.09 (m, 4*H*), 2.62 (t, *J* = 7.5 Hz, 2*H*), 2.73–2.83 (m, 4*H*), 5.31–5.44 (m, 6*H*), 6.82 (s, 1*H*), 7.15 (dd, *J*_1_ = 8.5 Hz, *J*_2_ = 2.0 Hz, 1*H*), 7.41 (d, *J* = 2.5 Hz, 1*H*), 7.50–7.56 (m, 3*H*), 7.90–7.91 (m, 2*H*), 8.24 (d, *J* = 8.5 Hz, 1*H*). ^13^C NMR (CDCl_3_, 125 MHz) δ (ppm): 14.3, 20.6, 24.8, 25.5, 25.6, 27.2, 29.0, 29.1, 29.1, 29.6, 34.4, 107.6, 111.1, 119.4, 121.7, 126.3 (2xC), 127.1, 127.1, 127.8, 128.2, 128.3, 129.1 (2xC), 130.2, 131.5, 131.7, 132.0, 154.7, 156.7, 163.6, 171.4, 177.7. HR-MS (ESI) *m/z* 521.2656 (calcd for C_33_H_38_O_4_Na [M + Na]^+^, 521.2668).

### 3.3. Biological Studies

In order to measure the antioxidant activity, the brains of untouched rats from the Central Laboratory of Experimental Animal (Medical University of Warsaw, Poland) were used. The brains were collected according to procedure no 3R. Human metastatic prostate cancer (PC3) and human immortal keratinocyte cell line from adult human skin (HaCaT) were used in the cytotoxicity studies.

#### 3.3.1. Cell Culture

Human prostate cancer cells (PC3) and human immortal keratinocyte cell line from adult human skin (HaCaT) were obtained from American Type Culture Collection (Teddington, UK) and cultured according to its instructions RPMI 1640 (L0490) medium or DMEM (L0102) medium, respectively, in a 95% air and 5% CO_2_ humidified incubator at 37 °C. Both media (Biowest) were supplemented with 10% fetal bovine serum (FBS), 100 U/mL penicillin and 100 μg/mL streptomycin. 10 mM stock solutions of tested compounds in DMSO were prepared.

#### 3.3.2. Viability Test

To assess the cells growth inhibition, the MTT dye reduction test was preformed based on the method described by Präbst et al. [66]. Both PC3 and HaCaT cells (1 × 10^3^/well) were seeded into 96-well plates and incubated for 24 h to attach to the bottom of the well in appropriate medium (200 μL) and treated with tested compounds (0–100 μM). The medium with compounds was removed after 72 h and MTT (0.5 mg/mL) in fresh growth medium without serum was added to each well. The plates were incubated in dark for 4 h at 37 °C. Then, the solution in each well was removed and refilled with 100 μL DMSO: isopropanol (1:1) to dissolve formazon crystals. The optical density was measured with an UVM 340 (ASYS Hitech GmbH, Austria) microplate reader at 570 nm. Replicates of three wells for each dosage, including vehicle control, were analyzed for each experiment. The experiments were conducted in triplicate. The results are expressed as IC_50_ values. The value of IC_50_ was estimated using CompuSyn version 1.0.

#### 3.3.3. Antioxidative Potential


**Brain preparation**


After preparation, the brain was placed on an ice plate and washed with cold 0.9% NaCl. 10% (m/v) brain solution was obtained by homogenization in cold 10 mM TRIS-HCl buffer, pH = 7.5. The homogenate was centrifuged (2000 rpm, 10 min, 4 °C) to obtain supernatant 1 (S1).


**Samples preparation**


100 µM stock solutions of test compounds in DMSO were prepared. The test compound solution was added to a 200 µM S1 homogenate to a final concentration of 10 µM.


**Estimation of lipid peroxidation.**


The oxidation reaction was induced by adding to the samples with or without the test compounds freshly prepared solutions: 0.1% FeCl_2_ (0.2 mL) and 1 mM ascorbic acid (0.4 mL) [59,67,68,69]. After incubation for 1 h at 37 °C, the reaction was stopped by adding 8.1% sodium dodecyl sulfate (SDS) solution. In order to determine the concentration of *MDA*, the acetic acid/HCl buffer at pH = 3.4 and 120 μL of 0.8% thiobarbituric acid (TBA) solution at pH = 6 were added to the samples. The samples were incubated for 1 h at 100 °C. After incubation, the samples were cooled on ice and centrifuged for 10 min at 4000 RPM. The absorbance was measured at a wavelength of 532 nm. Blank content: 0.5 mL of acetate buffer + 0.5 mL of TBA solution.

Calculation of malondialdehyde (*MDA*) concentration [68] (Equation (1)):(1)$$MDA\;\mathrm{concentration}=∆\;A_{532\;\mathrm{nm}}\;\times \;1.56\;\times 10^{-6}\;\left[\frac{\frac{\mathrm{nmol}\;MDA}{h}}{g\;\mathrm{tissue}}\right]$$

#### 3.3.4. Statistical Analyses

Statistical analyses performed using GraphPad Prism 9 software (GraphPad Software, San Diego, CA, USA). The results are expressed as mean ± SD from at least three independent experiments. The statistical significance of differences between means was established by ANOVA with Dunnett’s multiple comparison pos hoc test. *p* values below 0.05 were considered statistically significant.

## 4. Conclusions

All examined flavonoid conjugates exhibited an increased cytotoxic action towards HaCaT cells.

6-stearate and 7-linolenate conjugates (2 of 10 tested flavonoids derivatives) revealed no cytotoxic potential in PC3 cells.

These increases in the cytotoxicity are probably caused by an enhanced cell membrane crossing resulting from higher lipophilic properties of the conjugates.

The cytotoxic activity of all examined flavonoids esters strongly depends on the hydroxyl group position and the type of conjugated fatty acid.

We may conclude that the cytotoxic potential of the tested flavonoid derivatives is not specific toward prostate cancer cells because the derivatives also decrease the viability of normal cells.

It may be considered that a lower antioxidant potential of the derivatives examined may be caused by the esterification of OH groups in the flavonoids.

The obtained results will be used for further research to clarify why the esterification position and type of fatty acid involved are factors influencing the biological properties of flavonoids esters.

## Data Availability

Data is contained within the article or Appendix A.

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
