# Peer review of "Effect of Hydroxyl Groups Esterification with Fatty Acids on the Cytotoxicity and Antioxidant Activity of Flavones"

_molecules, 2022, doi:10.3390/molecules27020420_

Round 1

Reviewer 1 Report

In my opinion, studies on the activity of synthetic fatty acid - flavonoids conjugates are important. The presented research was carried out for the first time and involved studies of the biological activity of a high number of novel flavonoid conjugates. 

The obtained results are of interest in both practical and theoretical terms. However, many references are not styled according to the rules of Molecules. Except this drawback, I do not find other noticeable shortcomings and suggest that the paper can be accepted after minor revision.

Author Response

Dear Reviewer, thank you for your revision. 

The manuscript was corrected according to the your suggestions. 

Kind regards, 

Reviewer 2 Report

The authors reported in their study the effect of conjugation of flavonoids in the ring A at 6 or 7 position with fatty acids increases the hydrophobic properties of flavonoids and facilitates cellular absorption, but cytotoxic and biological activities of all examined compounds strongly depend on hydroxyl group position and fatty acid type. The manuscript is novel, well-written, presentes a good posibility for the enhancement of the therapeutical potential of flavonoids in the treatment of various human diseases. However, I have some suggestions to point out to the authors.

1- Add chloroplast to the sites of ROS generation at L36.

2- Add reference to L39 

3- The introduction section concentrate mainly on the ROS generation and effects, these information are well-established before and these redundant information add nothing to the manuscript value. I suggest to reduce the deundant parts, particularly L33-62.

4- Pleaae rephrase the sentence in lines 79-82. 

5- The aim of the work at the end of the introduction section should be improved to facilate its understanding to the reader.

6- The whole manuscript needs deep revision for typographic errors, particularly results section.

7- L408-410: no need for this introduction.

8- Please provide suitable references to the testes you made in Materials and Methods section.

9- L123: Additional information ......................... Remove this phrase.

10- L148: 7-hydroxyflavanone must be corrected into 7-hydroxy flavone.

11- Heading of Table 1 is incomplete, and the text below the table must be revised.

12- L168: Remove FA.

13- L213: Correct ROSs into ROS.

14- L213-220: the paragraph must be rephrased as it sounds non expressive.

15- L239: Masek et all is incorrectely cited in the text

16- L242: Wang is incorrectely cited in the text

17- L466: Revise. Which thesis?

18- Conclusion section can be concentrated based on the study findings without the repetation of the results. Also, it could include a future prospective on the use of flavonoids conjugates with fatty acids in treating various diseases based on the unique properties of these conjugates.

Author Response

Dear Reviewer, Thank you for your revision. We corrected the Manuscript according to your suggestion. Please see the attachment.

Kind regards,

Reviewer 3 Report

“Flavonoids and polyunsaturated fatty acids are recommended as potential therapeutical substances in the prevention or treatment of diseases associated with oxidative stress.” The efficacy in vivo of flavonoids is quite questionable, therefore the term recommended could be deceptive.

“Considering these studies,” which studies?

When referring to esters the authors should use the suffix -ate instead of -ic (e.g. linolenate not linolenic, sorbate not sorbic)

The introduction starts with a part (1/3) about ROS but in the work the radical scavenging is not a fundamental part.

 The remaining part of introduction cites many reviews that are based on studies performed exclusively on cell lines but still supporting claims as “Numerous studies have recommended flavonoids as potential substance not only in cancer prevention but also in cancer treatment”, to present there are substantially no flavonoids based therapies or conclusive studies demonstrating how flavonoids can exert specific functions in concentrations that can be achieved by the diet. One of the references cited states: “Flavonoids have been found to exert cytotoxic activities only at relatively high doses, within the micromolar concentration range. […] achieving the plasma levels sufficient to reveal antiproliferative and cytotoxic effects may not be possible via oral administration”. The serious problem of bioavailability is briefly mentioned by the authors but the state of the art, i.e. studies on flavonoids derivatives (ethers, esters, acetals, carbamates, others), is not appropriately described.

The cytotoxic activity reported in Table 1 is expressed only at high micromolar concentrations (millimolar in the text) and it is not specific for prostate cancer cells compared to keratinocyte cells, there are no experimental data to explain the mechanism of action and the results with different derivatives are quite scattered, the authors could not explain a trend.

The antioxidant power of flavonoid derivatives (three of them were not tested, why?) are quite low and their differences are not explained, why sorbate should exert its effect only in 4a and not in 5a?

Are compounds 1,2, 4a-e, 5a-e soluble in water at the concentrations used for the experiments? Are the esters stable in the cell or are they hydrolyzed as soon as they penetrate the cells?

In summary, the effects as chemotherapics and antioxidants of studied compounds are not important enough to justify a publication. The study will be relevant if the mechanism of action of the compounds and/or the effect of conjugation with different fatty acids was demonstrated and explained.

Author Response

Dear Reviewer, Thank you for your revision, please see the attachment.

Kind regards

Reviewer 4 Report

Grażyna et al., report “In this Effect of Hydroxyl Groups Esterification with Fatty Acids on 2 the Cytotoxicity and Antioxidant Activity of Flavones” Manuscript needs revision as suggested below:

1          What are PC3 and   HaCaT cells? When mentioning first time complete name should be   written.

2          Abstract should be revised and results should be discussed precisely.

3          Page 3: b-lactamases should be corrected as β-lactamases

4          oC should be corrected as oC

5          For better understanding, authors should provide 1H-NMR, C13-NMR and HR-MS   spectra in the supplementary materials.

6          Why Authors chose just PC3 for cytotoxicity? Justify.

7          What was the positive control and negative control in cytotoxicity studies? It should be    included.

8          Testing of cytotoxicity for 72 h is too long. How it can be justified? Experiments at           different times such as 24 and 48 h should be conducted and should be presented in a            comparison.

9          In the experimental section, authors stated that they have tested compounds in the  concentration (0-100 µM) while in the discussion it is written tested concentrations (5- 100 mM). Which one is correct?

10        Table 2 should be presented in a better way.

11        Authors should take care of spacing and alignments in the manuscript.

Author Response

Dear Reviewer, thank you for your revision. Please see the attachment.

Kind regards

Round 2

Reviewer 3 Report

By reading the reference cited by the author, Haddad AQ, Venkateswaran V, Viswanathan L, Teahan SJ, Fleshner NE, Klotz LH. Novel antiproliferative flavonoids induce cell cycle arrest in human prostate cancer cell lines. Prostate Cancer Prostatic Dis. 2006;9(1):68-76, it can be found that Haddaq et al. cite a study by Ganry (https://doi.org/10.1016/j.ypmed.2004.10.022 ) that in turn write: “Overall, the results of these studies do not show protective effects. Only four of these studies are prospective, and none of them found statistically significant prostate cancer reductions. Two prospective studies measured flavonoid intake and one reported a preventive effect on prostate cancer for the assumption of myricetin. One study assessed enterolactone concentrations in three different countries and showed no reduction in prostate cancer occurrence.” I advise the authors to be more cautious with “therapeutic” claiming.

Appropriate citation of polyphenols derivatives is still lacking, e.g.:

https://dx.doi.org/10.2478/intox-2013-0031

https://doi.org/10.1016/j.foodchem.2010.06.059

https://doi.org/10.1016/s0024-3205(99)00418-x

https://doi.org/10.1016/S1388-1981(99)00062-1

https://doi.org/10.1021/jm00352a019

https://doi.org/10.1007/s11033-021-06516-5

https://doi.org/10.1016/j.cbi.2020.109218

https://doi.org/10.3390/molecules191015900

https://doi.org/10.2174/1386207319666160202120928

The study from Bratton et al. (ref 41) is not so relevant with the present manuscript, since Bratton et al. used free fatty acids (linoleic, gamma-linolenic, arachidonic, alpha-linolenic, eicosapentaenoic, docosahexaenoic) and it can be seen that linoleic acid is less effective on PC3 cells than all the other tested fatty acids. In the manuscript, the most active compound is 7-hydroxy-flavone stearate, a derivative with a saturated fatty acid chain: this shows that unsaturation may not be a factor, in disagreement with reference 41.

Cytotoxic activity. The solubility of tested compounds is not an issue to be ignored: by adding 10 millimolar solution in DMSO of tested compound to a 200 microliter of medium, in order to obtain a 50 micromolar solution, you should add 1 microliter of DMSO with an amount of 3 – 6 micrograms of compound: visual assessment of turbidity could be deceptive. The detection of turbidity should be assessed by Dinamic Light Scattering. If the larger amount of DMSO added was 1%, the negative control should be 1%. All MTT assay results should be reported in the supplementary information to evaluate if there is a dose dependent effect.

“Changing the position of the substituent from 6 (4a) to 7 (5a) may cause a difference in the interaction of the sorbic acid chain with the electrons of the aromatic flavone ring. Such a change may result in differences in the biological activity of the derivative 4a and 5a.” This hypothesis must be verified by simple and fast measure of UV-visible spectra of compounds 4a-e and 5a-e.

Conclusions must be consistent with experimental results:

  1. “The conjugation increased the cytotoxic effects of the test flavonoids in the PC3 and HaCaT”: this is always true only for HaCaT.
  2. “This increase in the cytotoxicity probably resulted from an enhanced cell membrane crossing and higher concentrations of the flavonoids derivatives within the cell”: if this is true there should be some sort of trend in cytotoxicity.
  3. “It may be considered that further studies focused on searching flavonoid derivatives with better bioavailability”: this sentence doesn’t make sense.

Author Response

Dear Reviewer, thank you for your suggestion. We corrected manuscript according to your suggestions. Please see the attachment.

Kind regards

Reviewer 4 Report

Authors have revised the manuscript in the light of comments and responded to the comments. Therefore I suggest acceptance of this manuscript.

Author Response

Dear Reviewer, thank you for accepting our  revised manuscript.

kind regards